# Perceived Barriers and Facilitators in Cardiovascular Risk Management in Colombia: A Qualitative Analysis of the RE-HOPE Study

**DOI:** 10.3390/ijerph22081199

**Published:** 2025-07-31

**Authors:** Jose P. Lopez-Lopez, Yesica Giraldo-Castrillon, Johanna Otero, Claudia Torres, Alvaro Castañeda-Hernandez, Daniel Martinez-Bello, Claudia Garcia, Marianne Lopez-Cabrera, Patricio Lopez-Jaramillo

**Affiliations:** 1Masira Research Institute, Universidad de Santander (UDES), Bloque G, piso 6, Bucaramanga 680003, Colombia; josepatriciolopez@gmail.com (J.P.L.-L.); epidemioymetc.masira@udes.edu.co (Y.G.-C.); johanna.otero.w@gmail.com (J.O.); clau.torres@mail.udes.edu.co (C.T.); al.castaneda@mail.udes.edu.co (A.C.-H.); dan.martinez@mail.udes.edu.co (D.M.-B.); claudia.garcia@mail.udes.edu.co (C.G.); mariannelopez15@hotmail.com (M.L.-C.); 2Faculty of Dentistry, Universidad Santo Tomás, Bucaramanga 680006, Colombia

**Keywords:** cardiovascular risk, hypertension, barriers, facilitators, community-based health programs, qualitative study

## Abstract

**Introduction:** Low medication adherence and low hypertension control are a public health challenge, particularly in low- and middle-income countries (LMICs). Healthcare system- and patient-related barriers hinder the successful management of hypertension. This study aimed to identify the perceptions of barriers and facilitators to hypertension management among health system stakeholders in Santander, Colombia. **Materials and Methods:** We conducted a qualitative, phenomenological, and interpretative study, comprising five focus groups, to explore the barriers and facilitators to managing people with hypertension. Each focus group was formed by stakeholders from territorial entities, healthcare insurers, or healthcare providers. Meetings were held between December 2022 and February 2023. The sessions were recorded and transcribed using NVivo Transcription and analyzed using NVivo version 1.6.1. **Results:** Seven categories of barriers and facilitators were identified: strategies, resources, access, risk assessment, cross-sector collaboration, articulation, and stewardship. Of these categories, articulation and stewardship emerged as the main barriers, as revealed through axial coding and cluster analysis, which highlighted deficiencies in stewardship practices, a lack of clear objectives, and misalignment with public policy frameworks. **Conclusions:** Multisectoral actions extending beyond healthcare providers and aimed at improving coordination and intersectoral collaboration are essential for enhancing hypertension control in LMICs, such as Colombia. Addressing social determinants and strengthening primary healthcare through community-based strategies are critical, making stewardship and improved access key priorities.

## 1. Introduction

Globally, nearly one-third of adults are affected by hypertension, with the highest burden observed in low- and middle-income countries (LMICs) [1]. Despite advancements in pharmacological treatments and health services, less than 20% of individuals with hypertension achieve adequate control, likely due to barriers at the system, provider, and individual levels [2]. A systematic review of 28 studies identified key barriers to hypertension control, including limited time and attention from providers, ineffective interventions, and inadequate provider skills and attitudes [3]. More recently, an analysis of the HEARTS initiative for hypertension and cardiovascular risk management, which assessed 22 countries in the Americas, revealed significant barriers to accessing essential and optimal medications [4]. Conversely, several facilitators have been identified that support the implementation of relevant, consensus-driven, and sustainable interventions, such as family support, motivation, and the incorporation of lifestyle modifications [3]. In Colombia, cardiovascular risk management programs are part of a national and multisectoral strategy involving several stakeholders with clearly defined responsibilities. The healthcare system consists of three primary stakeholders: (i) the government, represented by national, departmental, and municipal health authorities or territorial entities, which supervise the accountability of healthcare insurers; (ii) healthcare insurers, responsible for ensuring population access to care; and (iii) healthcare providers, directly responsible for delivering care. Despite the efforts of each of these stakeholders, hypertension and cardiovascular risk control rates remain suboptimal [5]. Most existing data on barriers and facilitators to hypertension management come from high-income countries, where socioeconomic and health system contexts differ substantially [3]. In addition, most analyses of health system barriers have focused primarily on patients and healthcare providers, often overlooking the role of other key actors within the system (healthcare insurers or governmental representatives) [6]. Thus, there is a need to better understand the organizational, structural, and social determinants that affect hypertension control in LMICs like Colombia. Qualitative research offers valuable insights into the motivations, behaviors, and sociocultural contexts associated with the occurrence of health phenomena, facilitating the identification of programs, strategies, or approaches that are most effective. This study aimed to determine the perceptions of barriers and facilitators to managing hypertension among health system stakeholders in Santander, Colombia.

## 2. Materials and Methods

### 2.1. Study Type and Population

As part of the “Implementation, Integration, and Institutionalization of a Community-Based Care Program to Reduce Cardiovascular Risk in Santander” (RE-HOPE) implementation study, we conducted a qualitative, phenomenological study within an interpretive paradigm [7]. The RE-HOPE project aims to implement, integrate, and institutionalize a community-based care strategy to improve hypertension control across several municipalities in Santander, Colombia. RE-HOPE comprises two sequential phases: a preliminary qualitative phase aimed at identifying barriers and facilitators to hypertension control to guide intervention design, followed by an implementation phase to evaluate the effectiveness of the proposed strategy. The intervention includes early detection of hypertension, linkage to primary care services, and follow-up through home visits. For this qualitative analysis, we included healthcare providers from public and private institutions with expertise in service regulation, access, administration, and health insurance management. Participants were recruited from 11 public healthcare centers in Bucaramanga, Santander. This report follows the Consolidated Criteria for Reporting Qualitative Research (COREQ) guidelines [8].

### 2.2. Sample Selection

A typical case sampling approach was used to ensure a heterogeneous and representative group [9]. Focus groups were formed based on the following inclusion criteria: physicians, nurses, or nursing assistants with at least two years of experience and formal employment in a relevant institution—municipal or departmental health secretariats, healthcare insurers, or healthcare provider institutions. In line with standard recommendations, each focus group aimed to include between 6 and 10 participants to ensure optimal engagement [9].

### 2.3. Instruments and Data Collection Process

Eligible professionals were contacted by telephone and invited to voluntarily participate in the study. Upon agreement and the provision of informed consent, focus group sessions were scheduled between December 2022 and February 2023. The consent process included authorization for the recording and use of audio, photographs, and videos. Sessions were conducted in quiet, distraction-free environments at the participants’ workplaces to ensure fluid conversation and high-quality audio recordings. Research staff received training in standardized focus group facilitation and moderation techniques [9]. Each session was moderated by a member of the research team, while a second team member served as an observer. The moderator used a semi-structured interview guide and a predefined script, while the observer ensured adherence to protocol, documented nonverbal cues, and supported the overall process. Both team members facilitated active participation, ensuring that all attendees had the opportunity to express their views. Interviews were conducted in Spanish and later translated into English.

### 2.4. Processing and Analysis

Verbatim and automated transcriptions were generated using the NVivo Transcription system (NVivo software, version 1.6.1). Methodological rigor was ensured by applying Lincoln and Guba’s criteria for qualitative research (Table 1), which safeguarded against the influence of participants’ preexisting perceptions on the interpretation of their responses [10]. To preserve credibility and confidentiality, each recording was anonymized and assigned a unique code. The participants validated the findings during follow-up meetings. To ensure auditability, all textual reports were reviewed by both the session moderator and the observer, followed by a third independent review conducted by another member of the research team. The accuracy of the transcriptions was thoroughly verified. Exhaustiveness was achieved by applying the data saturation criterion. Saturation was determined after data collection. Two team members conducted an open comparison of categories by reading and rereading the collected information, and inter-coder agreement was established through a consensus process. Lastly, transferability was enhanced by tailoring context-specific questions to each participant’s lived experience. The qualitative analysis followed a multi-stage approach. A member of the research team who was not involved in the focus groups reviewed the transcripts and interpreted participants’ perceptions of barriers and facilitators to hypertension control. Based on participants’ experiences and perspectives, open coding was applied to develop initial thematic categories. Subsequently, an interpretive analysis was performed using axial coding to connect categories and organize narratives and accounts. Cluster analysis was then conducted to identify relationships within the textual data that could further enrich the understanding of perceived barriers and facilitators. In the final stage, confirmability was ensured through an independent review by a researcher not involved in data collection. As part of the audit trail, all qualitative analysis codes were archived in a Mendeley Data repository. The data collection protocol was approved by the Bioethics Committee of the Universidad de Santander (Act No. 018, dated 29 August 2022). The study was funded by Colombia’s General System of Royalties under BPIN code 2020000100447.

## 3. Results

A total of 25 representatives from the three main stakeholder groups—territorial entities, healthcare insurers, and healthcare providers—participated in this study. An average response rate of 89% was achieved across the five focus groups. Each session lasted between 19 and 57 min, and participants attended only one focus group. Most participants were women (84%), and more than one-third (80%) of the focus groups were conducted with individuals from territorial entities and healthcare insurers (Table 2). Each focus group consisted of an average of five participants (ranging from a minimum of three to a maximum of eight). Focus groups were stratified by the type of stakeholder, as territorial entities, health insurers, and healthcare providers represent the core components of the Colombian health system (Figure 1). During the open coding phase (descriptive analysis), seven key categories or codes were identified: strategies, resources, access, risk assessment, intersectoral collaboration, articulation, and leadership. These categories were derived by interpreting the transcripts within the broader context of public health practice in Colombia [11]. Definitions for each code are shown in Appendix A.

The distribution of topics or codes by stakeholder is presented in Table 3. Notably, the healthcare insurers did not reference codes such as access, intersectoral collaboration, or stewardship; the latter was exclusively mentioned by stakeholders from territorial entities. These participants also emphasized the importance of risk measurement, followed by intersectoral collaboration. In contrast, healthcare providers did not refer to risk measurement or stewardship. Their narratives predominantly focused on access, followed by intersectoral collaboration, and less frequently mentioned articulation or resources.

### 3.1. Findings by Category (Interpretative Analysis)

“Strategies” was the most frequently mentioned category in both the testimonies and the coding process. Stakeholders from insurance companies and territorial entities described different actions and plans aimed at identifying and following up with individuals at high risk for cardiovascular disease. These included efforts in health education and information dissemination to encourage engagement with healthcare providers. Community-based initiatives were also highlighted as key tools for raising public awareness. Several participants shared successful experiences involving the implementation and monitoring of strategies in collaboration with national and international organizations, which contributed to capacity building and strengthened intersectoral coordination within the department.

“In previous years, institutional strategies have focused on community engagement through health education initiatives. The departmental government implements targeted activities to ensure effective outreach at the municipal level.” Woman, 41, territorial entity group.

Stakeholders involved in population-level cardiovascular risk management recognized fragmentation across primary, secondary, and tertiary care levels. This fragmentation is characterized by difficulties in transferring information about high-risk populations, underscoring the need for improved coordination and more-integrated approaches. The negative impact of these barriers is exacerbated not only by the lack of community-based interventions but also by the failure of existing efforts to align with the objectives of the Collective Intervention Plans (PIC) implemented in the territories. Participants emphasized the crucial role of community leaders in implementing effective health strategies. They underlined the importance of enhancing information flow between entities, particularly from healthcare insurers to health providers. This concern was reflected in the focus group discussions as follows:

“Community leader engagement through loudspeaker announcements during health brigades represents a potentially effective strategy for health promotion outreach in this setting.” Woman, 26 years old, healthcare provider group.

Health provider and territorial entity stakeholders referred to the importance of implementing more effective strategies to reduce cardiovascular risk, particularly in individuals who resume healthy lifestyles and habits, combined with community-based actions, as shown below:

“Community leaders could enhance health outreach by utilizing public address systems during mobile health brigades to ensure broad dissemination of critical health information.” Woman, 31, healthcare insurer g.

Within the “resources” category, stakeholders from insurance companies highlighted their responsibility of managing resources at the individual level, which poses barriers due to the need for effective optimization and allocation across diverse territories. In contrast, resources at the population level are managed by territorial entities. Territorial entities support health promotion activities, risk control, and demand generation. Persistent resource shortages constitute a major barrier to inter-institutional collaboration. This often results in misalignments, as some stakeholders focus on individualized care while others prioritize collective interventions—each shaped by the imperative to respond to population health needs.

“A critical factor is the optimization of the Per Capita Payment Unit (UPC), which is the fixed budget allocated by the Ministry for population health risk management. We must maximize this limited UPC allocation; it’s not a matter of preferential treatment but rather operating within constrained resources that require systemic adaptation.” Woman, 39, healthcare insurer group.

“Articulation” was the third most frequently discussed category across the focus groups. Its importance was widely acknowledged; yet participants also emphasized its fragility among territorial entities, healthcare insurers, and both public and private healthcare providers at primary and secondary levels. Weak articulation among stakeholders was identified as a central barrier affecting the delivery of comprehensive care within cardiovascular risk management programs:

“Effective care integration requires strong coordination between providers. This coordination role falls to insurers through contractual agreements, we don’t contract individual providers, but rather comprehensive care pathways. However, aligning public and private networks within this framework presents significant operational challenges.” Woman, 39, healthcare insurer group.

“Better synchronization with healthcare insurers is necessary. Currently, while we successfully recruit patients and provide basic services like labs and medications, but many are lost before reaching specialists. The internal medicine referral package often fails when appointments aren’t secured, leading to patient attrition and fragmented care.” Woman, 36, healthcare provider group.

Several successful experiences of coordination between territorial entities, healthcare insurers, and healthcare providers, as well as between insurers and providers, were described. These cases demonstrated the potential to facilitate interaction, dialogue, and feedback across primary and complementary care services through professionals or analysts specifically assigned to the program. The importance of information documented in clinical records was reaffirmed, particularly regarding indicators used to monitor risk management (precursor conditions) and program adherence. These data serve as valuable inputs for evidence-based decision-making, including the development of comprehensive care pathways and strategies to reduce service fragmentation.

“The collective intervention plan and service provision framework have successfully integrated activities across municipal entities, territorial providers, and insurance organizations.” Woman, 41, territorial entity group.

“Access” to services was conceptualized as the actual potential for entry into and the utilization of essential healthcare services, based on insightful interpretations offered by institutional stakeholders. Overall, the narratives highlighted access barriers related to sociodemographic characteristics, limitations in social support networks, and logistical challenges faced by service providers, particularly in the distribution of medications for cardiovascular risk management.

“Patients from rural and remote areas face particularly severe challenges. The current individual care plans, including health promotion, maintenance, and complementary interventions, are insufficient given the geographic dispersion and resource constraints we experience in these regions.” Woman, 41, territorial entity group.

“Cross-sector collaboration” emerged as a critical category requiring coordinated efforts across the health system to effectively overcome key barriers and ensure the delivery of comprehensive care and services. Participants identified several challenges, including a lack of alignment among stakeholders, the absence of governance frameworks for resource management and allocation, and limited institutional capacity to coordinate and implement the diverse actions necessary to achieve integrated and effective care.

“Current efforts remain insufficient, as effective cardiovascular risk reduction requires coordinated actions across multiple sectors, including industry, education, and beyond,” Woman, 41, territorial entity group.

In the “risk measurement” category, the role of data flow and information exchange among system stakeholders was highlighted as essential for planning both individual- and population-level activities. Participants emphasized the importance of validating and enriching this information during the collection and analysis processes. Testimonies from representatives of territorial entities and healthcare insurers underscored the value placed on strengthening information systems to support continuous monitoring and evaluation of cardiovascular risk management at both individual and collective levels.

“The cardiovascular risk database comprehensively tracks all care pathway interventions, like taking the labs, imaging, and consults. This integrated system allows us to see how this evolution occurs.” Woman, 39, healthcare insurer group.

In contrast, some successful experiences were highlighted, including previous projects conducted in Santander that enabled the evaluation, analysis, and validation of cardiovascular risk measurement among key population stakeholders. For territorial entities, resuming such initiatives was deemed pertinent and potentially feasible through joint efforts and sustained implementation. However, healthcare providers did not offer specific insights or narratives related to this category.

“The STEPWise 1 and 2 studies represent successful initiatives that emerged from Santander’s sustained public health efforts. While these evidence-based programs have seen reduced support in recent years, their reactivation remains feasible through collective interventions. This approach wouldn’t require substantial new investments, but rather strategic reorganization of existing municipal resources.” Woman, 57, territorial entity group.

“Stewardship” was a category that emerged exclusively among stakeholders from territorial entities. Several barriers to effective state governance need to be addressed to foster collaboration and coordination in the implementation of public policies. While participants acknowledged the adequacy and clarity of existing legislation, regulations, and government strategies, they also emphasized a lack of personnel with the capacity to advocate for intersectoral agreements, monitor the organization of strategic plans, and manage cardiovascular risk and related interventions within the framework of the current Comprehensive Health Care Routes (RIAS).

“This initiative must be institutionalized as formal government policy to ensure consistency.” Woman, 41, territorial entity group.

The full quotes used for each category are shown in Appendix A.

### 3.2. Findings from the Cluster Analysis

The dendrogram derived from the cluster analysis (Figure 2) illustrates the set of emerging subcategories of patterns identified within some of the pre-established categories or codes. This analysis was conducted based on the similarity of words in the responses to questions posed by representatives of institutional stakeholders. The word “access” was always closely related to “cross-sector collaboration”, and, in turn, this was closely related to “strategies” and “articulation”. For access and cross-sector collaboration, subtopics such as difficulties, challenges, purposes, need for improvement, and effort in articulation were found; meanwhile, for articulation and strategies, difficulties and obstacles remained, along with implementation barriers. In the confirmability analysis, some interrelations between subcategories of interest were identified, demonstrating that barriers in access and the need for improvement were more closely related to health insurance. The relationship between the three types of stakeholders became evident in the subcategories of implementation barriers and purposes (Appendix A).

## 4. Discussion

In Colombia, the management of cardiovascular risk represents a complex challenge involving multiple institutional stakeholders within the healthcare system, including territorial entities, health insurers, and healthcare providers. This study identified the primary barriers to effective cardiovascular risk management to be a lack of robust strategies, limited resources, and inadequate inter-institutional coordination. Conversely, key facilitators included improved access to services, enhanced information flow, and strengthened stewardship, encompassing governance and leadership capacities. An interesting finding was the identification of categories that were strongly emphasized by some participants but entirely unaddressed by others—differences that appeared to be directly associated with the institutional roles of the entities they represented. For instance, stakeholders from healthcare insurers did not mention access, while those from healthcare providers did not address risk measurement; likewise, stewardship was exclusively discussed by representatives of territorial entities.

Generalizing the determinants of effective and contextually relevant management might be challenging. For instance, a study assessing the implementation of health system guidelines in LMICs showed that barriers and facilitators differ significantly depending on the type of institution and its specific functions within the system [12]. Additionally, accessibility and risk assessment are critical components for strengthening the resilience of health systems. In the present study, the codes regarding access mentioned by healthcare insurers and risk assessment mentioned by healthcare providers may indicate a disconnect between institutional public health policies and operational priorities [13]. Similarly, the concept of health system resilience highlights the importance of strong governance and stewardship as well as the active involvement of territorial entities in implementing health policies, which may explain why this category was mentioned exclusively by representatives from these institutions. Nevertheless, while stewardship is central to the mandate of territorial entities, its effective implementation also requires the collaboration and alignment of all stakeholders to ensure that such policies are not only adopted but also operationalized in accordance with each stakeholder’s institutional scope and capacity.

Several studies have examined the impact of information flow on the effectiveness of interventions in LMICs, and it has been reported that there is a significant need to improve articulation and equity in access to digital health information [3,14]. The perceived barriers centered on fragmentation between stakeholders, a lack of fluidity and relevance of information for decision-making, and the absence of clear purposes or goals aimed at achieving health objectives. For instance, in Latin American countries, it has been noted that fragmentation poses a significant challenge to scaling up effective secondary prevention strategies [15]. Ensuring that community-based strategies transcend the discourse of public policies toward their implementation is essential. The perceptions of providers regarding the characteristics of interventions to be implemented, as well as those of the recipients, are often overlooked. In this regard, the translation of evidence into practice influences the characteristics perceived by providers, impacting the efficiency of an implementation strategy [13]. Additionally, education strategies should be continued during the implementation of an intervention, given the demonstrated decline in knowledge retention among providers over time. Indeed, health literacy should involve both health professionals and non-medical personnel. The HOPE-4 study, conducted on 1371 hypertensive patients from Colombia and Malaysia, demonstrated that the implementation of a comprehensive care model led by trained non-medical health personnel, in conjunction with the continuous administration of appropriate pharmacological treatment and psychosocial support, increased pharmacological adherence in the intervention group and significantly reduced cardiovascular risk compared to standard care [16]. Nevertheless, a systematic review of both qualitative and quantitative studies found that most analyses of health system barriers have focused primarily on patients and healthcare providers, often overlooking the role of other key actors within the system (healthcare insurers or governmental representatives) [6]. Our findings offer a more comprehensive view by highlighting additional stakeholders involved in cardiovascular risk management in Colombia. This broader perspective supports the need for multisectoral interventions and emphasizes the importance of collaboration among primary care professionals, community organizations, and policymakers.

The lack of continuity appears to be associated with other key categories, including cross-sector collaboration, articulation, and stewardship. In this context, the importance of resources and effective stewardship in shaping public policy is particularly salient in low-resource settings [17]. Cross-sector collaboration (e.g., integrated health, education, and urban planning initiatives) and robust governance frameworks emerged as critical determinants for effective cardiovascular risk management. The implementation of these core components should be prioritized in population-level cardiovascular disease prevention strategies. Therefore, it is crucial to evaluate implementation outcomes, including acceptability, fidelity, feasibility, scalability, and sustainability, particularly in community-based strategies. However, evidence on the most appropriate tools to promote and support multisectoral actions remains limited. In LMICs, multisectoral approaches are often hindered by institutional weaknesses and fragmentation, including within the health sector itself, which undermines the coordination needed to address the structural drivers and social norms that impact vulnerable populations [18]. Addressing these multisectoral challenges requires robust governance and leadership; however, our findings reveal that leadership is frequently confined to territorial entities, with less-defined roles for other institutions and stakeholders. This was evidenced by the absence of references to governance responsibilities among health insurers and providers, as well as the prevailing perception that multisectoral coordination lies outside the health sector’s purview. Leadership must be collaborative and distributed across stakeholders, including non-health sectors such as education, sports, and culture, and institutional governance must be strengthened. Tackling social disparities requires directly addressing social determinants and aligning institutional mandates through coordinated, cross-sectoral efforts [18,19]. Practical recommendations include establishing territorial health councils with multisectoral representation, promoting community-based planning that incorporates vulnerable populations, and developing shared goals and performance indicators across sectors. Additionally, capacity-building programs for local officials and community leaders, along with intersectoral data-sharing platforms, can support evidence-informed decision-making and improve implementation fidelity.

## 5. Limitations

The projected participation rate was not obtained; however, the saturation of the categories was achieved with the representatives who participated. The male/female ratio of the recruited participants was heavily skewed, affecting the sample’s representativeness; however, sex is of lesser relevance here compared to patient focus groups. The absence of a follow-up questionnaire to quantify longitudinal changes in barriers and facilitators limits our ability to assess causality or temporal dynamics. The geographical scope of this study—limited to a single Colombian department and primarily urban participants—may not fully represent the diversity of the country’s health subsystems. This underscores the need to account for contextual factors to better understand the dynamics of interaction and power relations in each setting, and limits this study’s extrapolation to other areas of Colombia or other LMICs. However, we highlight the relevance of our findings, as there is limited evidence describing barriers and facilitators to cardiovascular risk management from the perspective of key stakeholders beyond patients and healthcare providers. Our study offers a novel perspective to address additional challenges in the effective management of cardiovascular risk.

## 6. Conclusions

Multisectoral interventions constitute a critical component of comprehensive cardiovascular risk reduction strategies, particularly in LMICs with high socioeconomic disparities. These interventions require decentralized leadership models that foster collaboration between primary care systems, community organizations, and policymakers. Despite evidence addressing governance and leadership barriers in managing specific health conditions or diseases, a notable gap remains in explicit research on conceptual frameworks and mechanisms related to stewardship from public health, administrative, or allied health fields. Critical components, such as cross-sector collaboration and stewardship, have not been recognized as fundamental elements, either as barriers or facilitators, in the effective management of cardiovascular risk. Studies should promote learning communities and evidence-based practices to strengthen multisectoral actions. Future research should explore conflicts of interest, organizational cultures, leadership mechanisms, and goal setting, while also assessing the transferability of results across institutions.

## Figures and Tables

**Figure 1 ijerph-22-01199-f001:**
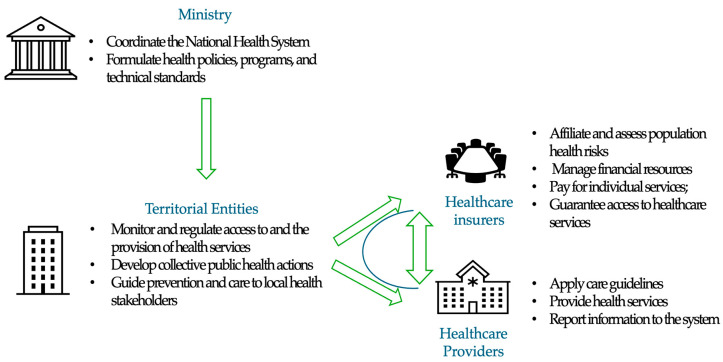
Primary stakeholders of the Colombian health system for cardiovascular risk management. The core components include territorial entities (supervising the accountability of healthcare insurers), health insurers (responsible for ensuring population access to care), and healthcare providers (directly accountable for delivering care).

**Figure 2 ijerph-22-01199-f002:**
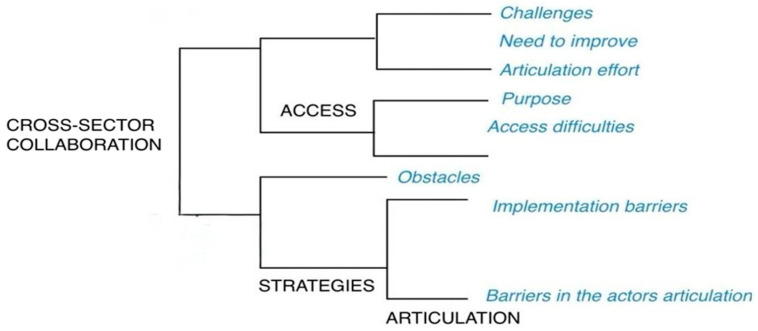
Dendrogram of the emerging subcategories derived from the cluster analysis by word similarity.

**Table 1 ijerph-22-01199-t001:** Question examples from the scripts provided to the focus group, categorized by stakeholder.

Stakeholder	Questions
Territorial entities	-According to your experience, what are the reasons that guarantee the management of cardiovascular risk? -How is access to the cardiovascular risk management program regulated? -How can the cardiovascular risk management be improved in Santander?
Healthcare insurers	-We would like to know in which way the activities between the entities and the community are articulated. -Describe how you manage the cardiovascular risk of your affiliates.
Healthcare providers	-Describe the follow up done to the participants of the cardiovascular risk programs. -What do you think could be the actions or community strategies to recruit and monitor patients? -In your opinion, what actions can be implemented to improve the cardiovascular risk program in your institution?

**Table 2 ijerph-22-01199-t002:** Focus group participants.

Stakeholder	Number of Focus Groups	Sex		Background
		Male	Female	
Territorial entities	2	2	12	Professional with specialty.
Healthcare insurers	2	2	3	Specialized professionals.
Healthcare providers	1	0	6	Doctors, nurses, and nursing assistants.

**Table 3 ijerph-22-01199-t003:** Distribution of codes or references identified by stakeholders in the focus groups.

Category	Proportion of Codes by Stakeholder		
	Healthcare Insurers (%) N = 5	Territorial Entities (%) N = 14	Healthcare Providers (%) N = 6
Access (n = 14)	0	21	79
Articulation (n = 17)	71	18	11
Strategies (n = 26)	54	27	19
Intersectorality (n = 6)	0	50	50
Risk Measurement (n = 6)	34	66	0
Stewardship (n = 3)	0	100	0
Resources (n = 20)	55	30	15

## Data Availability

Individual-level data will not be shared because RE-HOPE is an ongoing study. Requests for aggregated data will be considered on a case-by-case basis upon receipt of a reasonable request.

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
