# Peer review of "Perceived Barriers and Facilitators in Cardiovascular Risk Management in Colombia: A Qualitative Analysis of the RE-HOPE Study"

_ijerph, 2025, doi:10.3390/ijerph22081199_

Round 1
Reviewer 1 Report
Comments and Suggestions for Authors
The manuscript is concerned with a qualitative and interpretative study, utilising focus groups of private and public healthcare service providers, to explore the barriers and facilitators for managing people with hypertension in in low- and middle-income countries. The study was conducted in Columbia and involved 5 moderately, but sensibly sized focus groups.
The manuscript is largely very well written, but I struggled to fully understanding a number of the quotes of participants, probably due to the verbatim translation. The authors are encouraged to carefully review these, and, if necessary, moderate/adjust them to ‘clean verbatim’. Also, there are sections with quotes after quotas. It would be better to reduce these by extracting their essence instead.
The authors acknowledge not having obtained the projected participation rate, with numbers being only between 50% and 62.5% compared to anticipated. The male/female ratio of the recruited participants (4 males / 21 females) appears heavily skewed, though one may argue that sex is of lesser relevance here compared to patient focus groups. However, it is known that men have higher incidence as well as higher levels of hypertension in addition to lower levels of hypertension awareness than women. As for the latter point maybe some greater efforts could have been made to ensure a better gender balance.
Whilst the applied methodology is generally sound and appears to be have been well applied, it is a pity, in my view, that after having gathered a fair amount of qualitative information, apart from analysing the distribution of the codes of references, there was no attempt to quantify the information (e.g. by a follow-on questionnaire) for the work to be more impactful in terms of policy development.
Further, only one Colombian department was involved with the study with participants stemming mostly from urban areas. This raises the question as to how transferable the study is to other areas within Columbia, let alone other low- and middle-income countries.
Multisectoral actions were identified as being crucial for achieving optimal control of cardiovascular risk, a conclusion drawn also by other researchers. Particularly cross-sector collaboration and stewardship were identified as key facilitators, the latter in particular needs to be better explained.
Minor corrections:
- The abbreviations IPS, EAPB, ET and PIC appear to be based on the original Spanish words/phrases. Considering this is to be an international publication it would be better to base these on the English expressions instead.
- Line 25: Add a space in front of ‘Conclusion’
- Line 105: Add comma after ‘both’
- Line 126 (Table 1): Add the word ‘of’ after ‘participants’ in “…done to the participants the cardiovascular ….”
- Line 150: “X municipality” change to “municipality X”
- Line 173: “Santander en 173 Movimiento worked for about four, six years… for more than 10 years we worked first on 174 Carmen and then Santander en Movimiento.” Remove.
- Line 219: There are red quotation marks. Also “… is how 219 much the ministry gives me to be able to execute the risk.” is incomprehensible.
- Line 239: “we hire comprehensive routes”. What does that mean?
- Line 244: “internal medicine package”. What does that mean?
- Line 282: “The 281 difficulties there are many”. Improve English.
- Line 317: Add comma after “both”
- Line 373: The dendrogram is not particularly clear, e.g. why does cross-sector collaboration appear twice?
- Line 422: Add comma after “both”
Author Response
Comment #1: The manuscript is concerned with a qualitative and interpretative study, utilising focus groups of private and public healthcare service providers, to explore the barriers and facilitators for managing people with hypertension in in low- and middle-income countries. The study was conducted in Columbia and involved 5 moderately, but sensibly sized focus groups.
Authors: Thank you for the comment. We appreciate the reviewer for taking the time to review the manuscript carefully.
Comment #2: The manuscript is largely very well written, but I struggled to fully understanding a number of the quotes of participants, probably due to the verbatim translation. The authors are encouraged to carefully review these, and, if necessary, moderate/adjust them to ‘clean verbatim’. Also, there are sections with quotes after quotas. It would be better to reduce these by extracting their essence instead.
Authors: Thank you for the comment. We agree with the reviewer that a clear description of the quotes is necessary, and these have been added to the revised manuscript. In addition, we have included a Supplemental Table that extracts some of the quotes previously described in the main manuscript
Supplemental Table 2. Participants’ quotes for each category.
Code category |
Quotes |
Strategies |
· “Community groups previously employed trained health educators and physical activity facilitators. Our work began locally before scaling department-wide through the Santander en Movimiento program, which operated successfully for more than 10 years" Woman, 57, territorial entities. |
Stakeholders |
· "The patient risk identification process was initially implemented with a 2023 completion target, intended to generate consolidated data for insurers to develop evidence-based population intervention plans." Woman, 39, Healthcare Insurers. |
· "The EPS door-to-door initiative demonstrates particular value in our experience. While our facility faces challenges in patient retrieval, the most we can do is call, and many times it is lost, the insurer's home visits prove more effective, especially for elderly patients who respond better to printed appointment reminders and personal engagement." Woman, 50, healthcare providers. |
|
Community-based actions |
· "Health promotion efforts should prioritize lifestyle modifications, particularly through physical activity promotion and evidence-based nutritional guidance." Woman, 36, healthcare providers. |
Resources |
· "All activities within the collective intervention plan have been progressively implemented, particularly focusing on resource-limited municipalities. This initiative specifically supports early diagnosis of cardiovascular risk populations through complementary services.” Woman, 41, Territorial Entities. |
· "Dedicated personnel must be assigned to these activities. As healthcare providers, we already carry substantial clinical responsibilities with high time demands, making additional tasks unsustainable without proper staffing allocations." Woman, 41, Nurse, healthcare providers. |
|
Service fragmentation |
· “Our model's key advantage lies in its structured coordination mechanism: a dedicated cardiovascular risk professional within the healthcare insurers facilitates continuous collaboration between primary and complementary providers. This tripartite integration has improved care pathway efficiency, with measurable outcomes already emerging. Notably, our first-level training program from the complementary care component represents one of the most successful current initiatives for this patient cohort." Woman, 39, Healthcare Insurers. |
· “The IMAP (information system) team includes dedicated analysts and a supervising professional who track medication adherence and other key indicators. These monitoring reports inform our decisions as the contracting authority for providers." Woman, 36, Healthcare Insurers |
|
Access |
· "Our cardiovascular risk population is predominantly elderly, with limited education and minimal social support networks. These compounding factors, particularly low health literacy, are so significant that many patients don't even understand their prescribed medications." Woman, 52, Physician, healthcare providers. |
· "Frequent medication stockouts create dangerous care gaps: patients must return to physicians for alternative prescriptions, wait indefinitely for deliveries from the pharmacy, or ultimately disengage from care entirely. This systemic fragility particularly disadvantages vulnerable populations." Woman, 26 years old, Nursing Assistant, healthcare providers. |
|
Cross-sector collaboration |
· “Significant systemic challenges exist in coordinating with non-health sectors like infrastructure, sports, and education. A prime example is the CERS (Healthy Cities, Environments and Ruralities) strategy, its implementation requires collaboration on sports infrastructure that falls outside the health sector's jurisdiction, creating substantial interdepartmental coordination barriers." Woman, 57, territorial entities. |
· "While we recognize the need for substantial improvements in cardiovascular risk reduction, success ultimately depends on factors beyond our direct control, social determinants, and systemic conditions that collectively influence population health outcomes." Woman, 31, Nurse, healthcare providers. |
|
Risk measurement |
· "Program success is ultimately demonstrated through outcome indicators, these metrics provide the most objective evidence of intervention effectiveness." Woman, 39, territorial entities. |
Stewardship |
· “There are persistent systemic barriers at the departmental government level that hinder effective implementation." Woman, 57, territorial entities. |
· "We require dedicated personnel and policy clarity, not just more strategies. Territorial entities must prioritize and operationalize specific interventions through committed, focused action." Woman, 57, territorial entities. |
Comment #3: The authors acknowledge not having obtained the projected participation rate, with numbers being only between 50% and 62.5% compared to anticipated. The male/female ratio of the recruited participants (4 males / 21 females) appears heavily skewed, though one may argue that sex is of lesser relevance here compared to patient focus groups. However, it is known that men have higher incidence as well as higher levels of hypertension in addition to lower levels of hypertension awareness than women. As for the latter point maybe some greater efforts could have been made to ensure a better gender balance.
Authors: Thank you for the comment. We agree that, in this specific group of participants, sex differences are less relevant compared to general patient populations. We have addressed this limitation by adding the following text to our discussion:
- Limitations: The male/female ratio of the recruited participants is heavily skewed, affecting the sample’s representativeness; however, sex is of lesser relevance here compared to patient focus groups
Comment #4: Whilst the applied methodology is generally sound and appears to be have been well applied, it is a pity, in my view, that after having gathered a fair amount of qualitative information, apart from analysing the distribution of the codes of references, there was no attempt to quantify the information (e.g. by a follow-on questionnaire) for the work to be more impactful in terms of policy development.
Authors: Thank you for the recommendation. We will possibly consider it in our future research.
Comment #5: Further, only one Colombian department was involved with the study with participants stemming mostly from urban areas. This raises the question as to how transferable the study is to other areas within Columbia, let alone other low- and middle-income countries.
Authors: Thank you for the comment. We totally agree with the reviewer, and this issue was addressed in the limitations section of the manuscript.
- “The geographical scope of the study—limited to a single Colombian department and primarily urban participants—may not fully represent the diversity of the country’s health subsystems. This underscores the need to account for contextual factors to better understand the dynamics of interaction and power relations in each setting, and limits the study’s extrapolation to other areas of Colombia or other LMICs...”
Comment #6: Multisectoral actions were identified as being crucial for achieving optimal control of cardiovascular risk, a conclusion drawn also by other researchers. Particularly cross-sector collaboration and stewardship were identified as key facilitators, the latter in particular needs to be better explained.
Authors: This is a good point. To address this comment, we have added the following to the Discussion section of the revised manuscript:
- Cross-sector collaboration (e.g., integrated health, education, and urban planning initiatives) and robust governance frameworks emerged as critical determinants for effective cardiovascular risk management. Implementation of these core components should be prioritized in population-level cardiovascular disease prevention strategies.
Comment #7: The abbreviations IPS, EAPB, ET and PIC appear to be based on the original Spanish words/phrases. Considering this is to be an international publication it would be better to base these on the English expressions instead.
Authors: Thank you for the comment. As suggested by the reviewer, we have modified the original Spanish abbreviations.
Comment #8: Line 25: Add a space in front of ‘Conclusion’
Authors: Done.
Comment #9: Line 105: Add comma after ‘both’
Authors: Done.
Comment #10: Line 126 (Table 1): Add the word ‘of’ after ‘participants’ in “…done to the participants the cardiovascular ….”
Authors: Done.
Comment #11: Line 150: “X municipality” change to “municipality X”
Authors: Thank you for the comment. As suggested by the reviewer, we modified the expressions and decided to add a Supplemental Table that extracts some of the quotes previously described in the main manuscript.
Comment #12: Line 173: “Santander en 173 Movimiento worked for about four, six years… for more than 10 years we worked first on 174 Carmen and then Santander en Movimiento.” Remove.
Authors: Thank you for the comment. As suggested by the reviewer, we removed it and decided to add a Supplemental Table 2 that extracts some of the quotes previously described in the main manuscript.
Comment #13: Line 219: There are red quotation marks. Also “… is how 219 much the ministry gives me to be able to execute the risk.” is incomprehensible.
Authors: Thank you for the comment. As suggested by the reviewer, we have modified the expressions and decided to add a Supplemental Table 2 that extracts some of the quotes previously described in the main manuscript.
Comment #14: Line 239: “we hire comprehensive routes”. What does that mean?
Authors: Thank you for the comment. As suggested by the reviewer, we have modified the expressions. The original quote referred to the primary healthcare system’s order in Colombia. We also decided to add Supplemental Table 2, which extracts some of the quotes previously described in the main manuscript.
Comment #15: Line 244: “internal medicine package”. What does that mean?
Authors: Thank you for the comment. As suggested by the reviewer, we have modified the expressions. The original quote referred to prerequisite diagnostic tests (e.g., laboratory work, imaging) that patients must complete before specialist consultation. We also decided to add Supplemental Table 2, which extracts some of the quotes previously described in the main manuscript.
Comment #16: Line 282: “The 281 difficulties there are many”. Improve English.
Authors: Thank you for the comment. As suggested by the reviewer, we have modified the expressions and decided to add a Supplemental Table that extracts some of the quotes previously described in the main manuscript.
Comment #17: Line 317: Add comma after ‘both’
Authors: Done.
Comment #18: Line 373: The dendrogram is not particularly clear, e.g. why does cross-sector collaboration appear twice?
Authors: Thanks to the reviewer for pointing out this inconsistent issue. We have adjusted Figure 2 accordingly
Figure 2. Dendrogram of the emerging subcategories derived from the cluster analysis by word similarity.
Comment #19: Line 422: Add comma after ‘both’
Authors: Done.
Reviewer 2 Report
Comments and Suggestions for Authors
Brief summary: This qualitative study investigates perceived barriers and facilitators in the management of cardiovascular risk in Santander, Colombia, as part of the RE-HOPE implementation program. Using focus group discussions among stakeholders (health authorities, insurers, and providers), the authors identify seven main thematic categories: strategies, resources, access, risk assessment, intersectoral collaboration, articulation, and leadership. The study highlights challenges such as fragmented care, insufficient articulation across sectors, and resource constraints, while emphasizing community engagement, coordination, and cross-sectoral collaboration as critical enablers of improved care.
General Concept Comments: This is a well-executed qualitative study exploring a timely and under-researched topic in the context of LMICs. The use of the COREQ checklist and phenomenological approach is appropriate for the study's aims. The findings are relevant for informing health policy and implementation frameworks in cardiovascular risk management. However, while the manuscript is comprehensive, it would benefit from a more concise presentation of the findings and deeper critical reflection on their implications.
Specific comments:
Introduction:
The role of the RE-HOPE program is mentioned but could be more explicitly framed as part of the introduction’s conceptual foundation
Methods:
Focus group sizes:
Please clarify how many participants were included in each focus group (e.g., the minimum, maximum, and average number per group), as this helps assess the diversity and balance of views in each session.
Saturation:
It would be helpful to specify whether thematic saturation was planned in advance (prospectively) or determined after data collection (retrospectively), and how the research team judged that no new themes were emerging.
Results:
The section is thorough but quite long. Consider summarizing key findings from each category and moving detailed quotations to a supplemental table.
The use of stakeholder quotes is effective, but some repetition could be reduced and even summarized without transforming the exact words.
Gender distribution is extremely skewed (84% female); this should be acknowledged in the limitations section.
In Figure 1, key categories such as “Access” appear more than once in different branches of the dendrogram, which may be confusing to readers. Please consider simplifying or reorganizing the visual to avoid duplication or clearly explain why this repetition occurs.
Discussion:
The HOPE-4 study is a useful comparator. More contrast with similar qualitative efforts would strengthen the contextual relevance.
Some assertions could benefit from deeper interpretation (e.g., how stewardship or cross-sector barriers translate to practical obstacles for patients).
The last paragraph of the discussion is particularly strong and could be expanded with more specific implications for implementation science or policy development.
Comments on the Quality of English Language
Consider tightening the results narrative to reduce redundancy.
There are some long, complex sentences that could be broken down for clarity.
Define all acronyms at first use (e.g., IPS, EAPB, PIC, UPC).
Author Response
Reviewer 2
Comment #20: This is a well-executed qualitative study exploring a timely and under-researched topic in the context of LMICs. The use of the COREQ checklist and phenomenological approach is appropriate for the study's aims. The findings are relevant for informing health policy and implementation frameworks in cardiovascular risk management. However, while the manuscript is comprehensive, it would benefit from a more concise presentation of the findings and deeper critical reflection on their implications.
Authors: Thank you for the recommendation. We will possibly consider it in our future research.
Introduction
Comment #21: The role of the RE-HOPE program is mentioned but could be more explicitly framed as part of the introduction’s conceptual foundation.
Authors: Thank you for the recommendation. We agree with the reviewer, and the text has been added to the revised manuscript:
- “The RE-HOPE project aims to implement, integrate, and institutionalize a community-based care strategy to improve hypertension control across several municipalities in Santander, Colombia. RE-HOPE comprises two sequential phases: a preliminary qualitative phase aimed at identifying barriers and facilitators to hypertension control to guide the intervention design, followed by an implementation phase to evaluate the effectiveness of the proposed strategy. The intervention includes early detection of hypertension, linkage to primary care services, and follow-up through home visits.”
Methods
Comment #22: Focus group sizes: Please clarify how many participants were included in each focus group (e.g., the minimum, maximum, and average number per group), as this helps assess the diversity and balance of views in each session.
Authors: Thank you for your suggestion. We have added the following text to the Methods section:
- Each focus group consisted of an average of 5 participants (ranging from a minimum of 3 to a maximum of 8). Focus groups were stratified by type of stakeholder, as territorial entities, health insurers, and healthcare providers represent the core components of the Colombian health system (Figure 1).
Figure 1. Primary stakeholders of the Colombian health system for cardiovascular risk management. The core components include territorial entities (supervising the accountability of healthcare insurers), health insurers (responsible for ensuring population access to care), and healthcare providers (directly accountable for delivering care).
Comment #23: Saturation: It would be helpful to specify whether thematic saturation was planned in advance (prospectively) or determined after data collection (retrospectively), and how the research team judged that no new themes were emerging.
Authors: Thank you for your suggestion. We added the following text to the methods section of the revised manuscript.
- Saturation was determined after data collection. Two team members conducted an open comparison of categories by reading and rereading the collected information, and inter-coder agreement was established through a consensus process.
Results
Comment #24: The section is thorough but quite long. Consider summarizing key findings from each category and moving detailed quotations to a supplemental table.
Authors: Thank you for your suggestion. We agree with the reviewer and have added a Supplemental Table that extracts some of the quotes previously described in the main manuscript.
Supplemental Table 2. Participants’ quotes for each category.
Code category |
Quotes |
Strategies |
· “Community groups previously employed trained health educators and physical activity facilitators. Our work began locally before scaling department-wide through the Santander en Movimiento program, which operated successfully for more than 10 years" Woman, 57, territorial entities. |
Stakeholders |
· "The patient risk identification process was initially implemented with a 2023 completion target, intended to generate consolidated data for insurers to develop evidence-based population intervention plans." Woman, 39, Healthcare Insurers. |
· "The EPS door-to-door initiative demonstrates particular value in our experience. While our facility faces challenges in patient retrieval, the most we can do is call, and many times it is lost, the insurer's home visits prove more effective, especially for elderly patients who respond better to printed appointment reminders and personal engagement." Woman, 50, healthcare providers. |
|
Community-based actions |
· "Health promotion efforts should prioritize lifestyle modifications, particularly through physical activity promotion and evidence-based nutritional guidance." Woman, 36, healthcare providers. |
Resources |
· "All activities within the collective intervention plan have been progressively implemented, particularly focusing on resource-limited municipalities. This initiative specifically supports early diagnosis of cardiovascular risk populations through complementary services.” Woman, 41, Territorial Entities. |
· "Dedicated personnel must be assigned to these activities. As healthcare providers, we already carry substantial clinical responsibilities with high time demands, making additional tasks unsustainable without proper staffing allocations." Woman, 41, Nurse, healthcare providers. |
|
Service fragmentation |
· “Our model's key advantage lies in its structured coordination mechanism: a dedicated cardiovascular risk professional within the healthcare insurers facilitates continuous collaboration between primary and complementary providers. This tripartite integration has improved care pathway efficiency, with measurable outcomes already emerging. Notably, our first-level training program from the complementary care component represents one of the most successful current initiatives for this patient cohort." Woman, 39, Healthcare Insurers. |
· “The IMAP (information system) team includes dedicated analysts and a supervising professional who track medication adherence and other key indicators. These monitoring reports inform our decisions as the contracting authority for providers." Woman, 36, Healthcare Insurers |
|
Access |
· "Our cardiovascular risk population is predominantly elderly, with limited education and minimal social support networks. These compounding factors, particularly low health literacy, are so significant that many patients don't even understand their prescribed medications." Woman, 52, Physician, healthcare providers. |
· "Frequent medication stockouts create dangerous care gaps: patients must return to physicians for alternative prescriptions, wait indefinitely for deliveries from the pharmacy, or ultimately disengage from care entirely. This systemic fragility particularly disadvantages vulnerable populations." Woman, 26 years old, Nursing Assistant, healthcare providers. |
|
Cross-sector collaboration |
· “Significant systemic challenges exist in coordinating with non-health sectors like infrastructure, sports, and education. A prime example is the CERS (Healthy Cities, Environments and Ruralities) strategy, its implementation requires collaboration on sports infrastructure that falls outside the health sector's jurisdiction, creating substantial interdepartmental coordination barriers." Woman, 57, territorial entities. |
· "While we recognize the need for substantial improvements in cardiovascular risk reduction, success ultimately depends on factors beyond our direct control, social determinants, and systemic conditions that collectively influence population health outcomes." Woman, 31, Nurse, healthcare providers. |
|
Risk measurement |
· "Program success is ultimately demonstrated through outcome indicators, these metrics provide the most objective evidence of intervention effectiveness." Woman, 39, territorial entities. |
Stewardship |
· “There are persistent systemic barriers at the departmental government level that hinder effective implementation." Woman, 57, territorial entities. |
· "We require dedicated personnel and policy clarity, not just more strategies. Territorial entities must prioritize and operationalize specific interventions through committed, focused action." Woman, 57, territorial entities. |
Comment #25: The use of stakeholder quotes is effective, but some repetition could be reduced and even summarized without transforming the exact words.
Authors: Thank you for your suggestion. We have adjusted this section accordingly.
Comment #26: Gender distribution is extremely skewed (84% female); this should be acknowledged in the limitations section.
Authors: Thank you for the comment. We agree that in this specific group of participants, sex differences are less relevant compared to general patient populations. We have addressed this limitation by adding the following text to our discussion:
- Limitations: The male/female ratio of the recruited participants is heavily skewed, affecting the sample’s representativeness; however, sex is of lesser relevance here compared to patient focus groups.
Comment #27: In Figure 1, key categories such as “Access” appear more than once in different branches of the dendrogram, which may be confusing to readers. Please consider simplifying or reorganizing the visual to avoid duplication or clearly explain why this repetition occurs
Authors: Thanks to the reviewer for pointing out this inconsistent issue. We have adjusted Figure 2 accordingly.
Figure 2. Dendrogram of the emerging subcategories derived from the cluster analysis by word similarity.
Discussion
Comment #28: The HOPE-4 study is a useful comparator. More contrast with similar qualitative efforts would strengthen the contextual relevance.
Authors: Thank you for the recommendation. We agree with the reviewer, and these have been added to the discussion section in the revised manuscript.
- Nevertheless, a systematic review of both qualitative and quantitative studies found that most analyses of health system barriers have focused primarily on patients and healthcare providers, often overlooking the role of other key actors within the system (healthcare insurers or governmental representatives) (1). Our findings offer a more comprehensive view by highlighting additional stakeholders involved in cardiovascular risk management in Colombia. This broader perspective supports the need for multisectoral interventions and emphasizes the importance of collaboration among primary care professionals, community organizations, and policymakers.
Khatib R, Schwalm JD, Yusuf S, Haynes RB, McKee M, Khan M, et al. Patient and healthcare provider barriers to hypertension awareness, treatment and follow up: a systematic review and meta-analysis of qualitative and quantitative studies. PLoS One. 2014;9(1):e84238.
Comment #29: Some assertions could benefit from deeper interpretation (e.g., how stewardship or cross-sector barriers translate to practical obstacles for patients).
Authors: This is a good point. To address this comment, we have added the following to the Discussion section of the revised manuscript:
- Cross-sector collaboration (e.g., integrated health, education, and urban planning initiatives) and robust governance frameworks emerged as critical determinants for effective cardiovascular risk management. Implementation of these core components should be prioritized in population-level cardiovascular disease prevention strategies.
Comment #30: The last paragraph of the discussion is particularly strong and could be expanded with more specific implications for implementation science or policy development.
Authors: Thank you for your suggestion. We have adjusted this section accordingly.
- Practical recommendations include establishing territorial health councils with multisectoral representation, promoting community-based planning that incorporates vulnerable populations, and developing shared goals and performance indicators across sectors. Additionally, capacity-building programs for local officials and community leaders, along with intersectoral data-sharing platforms, can support evidence-informed decision-making and improve implementation fidelity.
Comment #31: Consider tightening the results narrative to reduce redundancy.
Authors: Thank you for the comment. We agree with the reviewer and have added a Supplemental Table 2 that extracts some of the quotes previously described in the main manuscript.
Supplemental Table 2. Participants’ quotes for each category.
Code category |
Quotes |
Strategies |
· “Community groups previously employed trained health educators and physical activity facilitators. Our work began locally before scaling department-wide through the Santander en Movimiento program, which operated successfully for more than 10 years" Woman, 57, territorial entities. |
Stakeholders |
· "The patient risk identification process was initially implemented with a 2023 completion target, intended to generate consolidated data for insurers to develop evidence-based population intervention plans." Woman, 39, Healthcare Insurers. |
· "The EPS door-to-door initiative demonstrates particular value in our experience. While our facility faces challenges in patient retrieval, the most we can do is call, and many times it is lost, the insurer's home visits prove more effective, especially for elderly patients who respond better to printed appointment reminders and personal engagement." Woman, 50, healthcare providers. |
|
Community-based actions |
· "Health promotion efforts should prioritize lifestyle modifications, particularly through physical activity promotion and evidence-based nutritional guidance." Woman, 36, healthcare providers. |
Resources |
· "All activities within the collective intervention plan have been progressively implemented, particularly focusing on resource-limited municipalities. This initiative specifically supports early diagnosis of cardiovascular risk populations through complementary services.” Woman, 41, Territorial Entities. |
· "Dedicated personnel must be assigned to these activities. As healthcare providers, we already carry substantial clinical responsibilities with high time demands, making additional tasks unsustainable without proper staffing allocations." Woman, 41, Nurse, healthcare providers. |
|
Service fragmentation |
· “Our model's key advantage lies in its structured coordination mechanism: a dedicated cardiovascular risk professional within the healthcare insurers facilitates continuous collaboration between primary and complementary providers. This tripartite integration has improved care pathway efficiency, with measurable outcomes already emerging. Notably, our first-level training program from the complementary care component represents one of the most successful current initiatives for this patient cohort." Woman, 39, Healthcare Insurers. |
· “The IMAP (information system) team includes dedicated analysts and a supervising professional who track medication adherence and other key indicators. These monitoring reports inform our decisions as the contracting authority for providers." Woman, 36, Healthcare Insurers |
|
Access |
· "Our cardiovascular risk population is predominantly elderly, with limited education and minimal social support networks. These compounding factors, particularly low health literacy, are so significant that many patients don't even understand their prescribed medications." Woman, 52, Physician, healthcare providers. |
· "Frequent medication stockouts create dangerous care gaps: patients must return to physicians for alternative prescriptions, wait indefinitely for deliveries from the pharmacy, or ultimately disengage from care entirely. This systemic fragility particularly disadvantages vulnerable populations." Woman, 26 years old, Nursing Assistant, healthcare providers. |
|
Cross-sector collaboration |
· “Significant systemic challenges exist in coordinating with non-health sectors like infrastructure, sports, and education. A prime example is the CERS (Healthy Cities, Environments and Ruralities) strategy, its implementation requires collaboration on sports infrastructure that falls outside the health sector's jurisdiction, creating substantial interdepartmental coordination barriers." Woman, 57, territorial entities. |
· "While we recognize the need for substantial improvements in cardiovascular risk reduction, success ultimately depends on factors beyond our direct control, social determinants, and systemic conditions that collectively influence population health outcomes." Woman, 31, Nurse, healthcare providers. |
|
Risk measurement |
· "Program success is ultimately demonstrated through outcome indicators, these metrics provide the most objective evidence of intervention effectiveness." Woman, 39, territorial entities. |
Stewardship |
· “There are persistent systemic barriers at the departmental government level that hinder effective implementation." Woman, 57, territorial entities. |
· "We require dedicated personnel and policy clarity, not just more strategies. Territorial entities must prioritize and operationalize specific interventions through committed, focused action." Woman, 57, territorial entities. |
Comment #32: There are some long, complex sentences that could be broken down for clarity.
Authors: Done.
Comment #33: Define all acronyms at first use (e.g., IPS, EAPB, PIC, UPC).
Authors: Thank you for the comment. As suggested by the reviewer, we have modified the original Spanish acronyms.
Reviewer 3 Report
Comments and Suggestions for Authors
The authors have conducted a qualitative study that addresses an important public health issue in LMICs: the barriers and facilitators of cardiovascular risk management from the perspective of system stakeholders. The use of focus groups, coding techniques and analytical triangulation (open, axial and cluster coding) adds robustness to the design and interpretative depth to the findings. However, improvements are recommended:
- The background is well structured and includes up-to-date references. However, line 57–63 would benefit from a clearer articulation of the knowledge gap addressed by the study.
- The methodology is quite comprehensive and transparent. The use of the COREQ checklist, coding strategy and audit trail increases the reliability of the study. A brief explanation of how inter-coder agreement was approached (e.g. kappa scores or consensus process) could be considered. Also, a clearer justification of the stratification of focus groups by stakeholder, beyond heterogeneity (lines 76-83).
- In the results section, Table 3 could be improved by including a total of N per stakeholder group. Figure 1 (dendrogram) needs a more complete legend.
-
In lines 403-407, the management argument could be reinforced with a brief table or visual element (e.g. a stakeholder role matrix) showing who takes what responsibilities.
-
Lines 420-427 could benefit from a brief restatement of how this study complements and contrasts with the HOPE-4 findings.
Comments on the Quality of English Language
The manuscript is well organized, but several sentences are long, redundant, or awkward. Examples:
-
-
Line 13–14: “the perceptions of barriers and facilitators for managing hypertension…”
-
Line 354–356: “many strategies… but get your teeth into it…”
-
Line 457–460: Consider rephrasing for more fluidity.
-
Author Response
Comment #34: The authors have conducted a qualitative study that addresses an important public health issue in LMICs: the barriers and facilitators of cardiovascular risk management from the perspective of system stakeholders. The use of focus groups, coding techniques and analytical triangulation (open, axial and cluster coding) adds robustness to the design and interpretative depth to the findings. However, improvements are recommended:
Authors: Thank you for the comment. We appreciate the reviewer for taking the time to review the manuscript carefully.
Comment #35: The background is well structured and includes up-to-date references. However, line 57–63 would benefit from a clearer articulation of the knowledge gap addressed by the study.
Authors: Thank you for the recommendation. We agree with the reviewer, and these have been added to the introduction section in the revised manuscript.
- In addition, most analyses of health system barriers have focused primarily on patients and healthcare providers, often overlooking the role of other key actors within the system (healthcare insurers or governmental representatives).
Comment #36: The methodology is quite comprehensive and transparent. The use of the COREQ checklist, coding strategy and audit trail increases the reliability of the study. A brief explanation of how inter-coder agreement was approached (e.g. kappa scores or consensus process) could be considered. Also, a clearer justification of the stratification of focus groups by stakeholder, beyond heterogeneity (lines 76-83).
Authors: Thank you for the comment. We added the following text to the methods section of the revised manuscript.
- Saturation was determined after data collection. Two team members conducted an open comparison of categories by reading and rereading the collected information, and inter-coder agreement was established through a consensus process.
- Focus groups were stratified by type of stakeholder, as territorial entities, health insurers, and healthcare providers represent the core components of the Colombian health system (Figure 1).
Comment #37: In the results section, Table 3 could be improved by including a total of N per stakeholder group. Figure 1 (dendrogram) needs a more complete legend.
Authors: Thanks to the reviewer for pointing out this detail. We have adjusted Table 3 and Figure 2 accordingly.
Table 3. Distribution of codes or references identified by stakeholders in the focus groups.
Category |
Proportion of codes by stakeholder |
||
Healthcare Insurers (%) n= 5 |
Territorial Entities (%) n= 14 |
Healthcare Providers (%) n= 6 |
|
Access (n= 14) |
0 |
21 |
79 |
Articulation (n= 17) |
71 |
18 |
11 |
Strategies (n= 26) |
54 |
27 |
19 |
Intersectorality (n= 6) |
0 |
50 |
50 |
Risk Measurement (n= 6) |
34 |
66 |
0 |
Stewardship (n= 3) |
0 |
100 |
0 |
Resources (n= 20) |
55 |
30 |
15 |
Figure 2. Dendrogram of the emerging subcategories derived from the cluster analysis by word similarity.
Comment #38: In lines 403-407, the management argument could be reinforced with a brief table or visual element (e.g. a stakeholder role matrix) showing who takes what responsibilities.
Authors: Thanks to the reviewer for pointing out this detail. We have adjusted and added a new Figure 1, seen in the text in the following way:
Figure 1. Primary stakeholders of the Colombian health system for cardiovascular risk management. The core components include territorial entities (supervising the accountability of healthcare insurers), health insurers (responsible for ensuring population access to care), and healthcare providers (directly accountable for delivering care).
Comment #39: Lines 420-427 could benefit from a brief restatement of how this study complements and contrasts with the HOPE-4 findings.
Authors: Thank you for the recommendation. We agree with the reviewer, and these have been added to the discussion section in the revised manuscript.
- Nevertheless, a systematic review of both qualitative and quantitative studies found that most analyses of health system barriers have focused primarily on patients and healthcare providers, often overlooking the role of other key actors within the system (healthcare insurers or governmental representatives) (1). Our findings offer a more comprehensive view by highlighting additional stakeholders involved in cardiovascular risk management in Colombia. This broader perspective supports the need for multisectoral interventions and emphasizes the importance of collaboration among primary care professionals, community organizations, and policymakers.
- Khatib R, Schwalm JD, Yusuf S, Haynes RB, McKee M, Khan M, et al. Patient and healthcare provider barriers to hypertension awareness, treatment and follow up: a systematic review and meta-analysis of qualitative and quantitative studies. PLoS One. 2014;9(1):e84238.
Comment #40: Line 13–14: “the perceptions of barriers and facilitators for managing hypertension…”
Authors: Thanks for the comment, we agree with the reviewer, and we have rephrased the sentence, as seen in the text, in the following way:
- “the perceptions of barriers and facilitators to hypertension management”.
Comment #41: Line 354–356: “many strategies… but get your teeth into it…”
Authors: Thank you for the comment. We agree with the reviewer that a clear description of the quotes is necessary, and these have been added to the revised manuscript. In addition, we have included a Supplemental Table 2 that extracts some of the quotes previously described in the main manuscript.
Comment #42: Line 457–460: Consider rephrasing for more fluidity.
Authors: Thanks for the comment, we agree with the reviewer, and we have rephrased the sentence, as seen in the text, in the following way:
- “Multisectoral interventions constitute a critical component of comprehensive cardiovascular risk reduction strategies, particularly in low-income and middle-income countries (LMICs) with high socioeconomic disparities. These interventions require decentralized leadership models that foster collaboration between primary care systems, community organizations, and policymakers.”
Round 2
Reviewer 2 Report
Comments and Suggestions for Authors
The authors have adequately addressed all the points raised in the previous round of review. I have no further comments or concerns.
Author Response
Thank you for the comment. We appreciate the reviewer for taking the time to review the manuscript carefully.